# Cell-Derived Exosomes as Therapeutic Strategies and Exosome-Derived microRNAs as Biomarkers for Traumatic Brain Injury

**DOI:** 10.3390/jcm11113223

**Published:** 2022-06-05

**Authors:** Jing Wang, Junwen Wang, Xinyan Li, Kai Shu

**Affiliations:** 1Department of Neurosurgery, Tongji Hospital, Tongji Medical College, Huazhong University of Science and Technology, Wuhan 430030, China; yuezhizhiye@163.com (J.W.); jwwang@tjh.tjmu.edu.cn (J.W.); 2Department of Neurosurgery, Third Hospital of Shanxi Medical University, Shanxi Bethune Hospital, Shanxi Academy of Medical Sciences, Tongji Shanxi Hospital, Taiyuan 030032, China; 3Department of Physiology, School of Basic Medicine, Tongji Medical College, Huazhong University of Science and Technology, Wuhan 430030, China

**Keywords:** traumatic brain injury, exosomes, extracellular vesicles, microRNAs

## Abstract

Traumatic brain injury (TBI) is a complex, life-threatening condition that causes mortality and disability worldwide. No effective treatment has been clinically verified to date. Achieving effective drug delivery across the blood–brain barrier (BBB) presents a major challenge to therapeutic drug development for TBI. Furthermore, the field of TBI biomarkers is rapidly developing to cope with the many aspects of TBI pathology and enhance clinical management of TBI. Exosomes (Exos) are endogenous extracellular vesicles (EVs) containing various biological materials, including lipids, proteins, microRNAs, and other nucleic acids. Compelling evidence exists that Exos, such as stem cell-derived Exos and even neuron or glial cell-derived Exos, are promising TBI treatment strategies because they pass through the BBB and have the potential to deliver molecules to target lesions. Meanwhile, Exos have decreased safety risks from intravenous injection or orthotopic transplantation of viable cells, such as microvascular occlusion or imbalanced growth of transplanted cells. These unique characteristics also create Exos contents, especially Exos-derived microRNAs, as appealing biomarkers in TBI. In this review, we explore the potential impact of cell-derived Exos and exosome-derived microRNAs on the diagnosis, therapy, and prognosis prediction of TBI. The associated challenges and opportunities are also discussed.

## 1. Introduction

Traumatic brain injury (TBI) is a common public health problem, and no effective clinical treatment strategies have been found; nearly half of the world’s population will experience one or more TBIs in their lifetime [1,2]. Despite rapid progress in understanding the pathophysiology of TBI, patients with moderate–severe TBI still suffer from long-term neurological deficits [3]. 

TBI is a multi-phase pathology with complex interactions between the brain, periphery, and immune system [4]. The disease triggers a series of complex pathological reactions that can be divided into two main stages: primary injury and secondary injury. Primary injury is caused by external impact, resulting in acute pathological changes, including brain contusion, intracerebral hemorrhage, and axonal shearing. Secondary pathological changes include oxidative stress, mitochondrial injury, glutamate excitotoxicity, Ca^2+^ overload, and neuroinflammation, leading to further deterioration of neurological functions [5]. Despite the remarkable advances in pathology, there are still no evidence-based therapies to control nerve damage following TBI [6], and the molecular events following TBI remain to be fully elucidated.

The family of extracellular vesicles (EVs) comprises a heterogeneous mixture of exosomes (Exos, 10–100 nm diameter), microvesicles (MVs) (also called microparticles (MPs), 100–1000 nm diameter), and apoptotic bodies (1–5 nm diameter) [7]. EVs may play a dual role in cells: On one hand, EVs may be received from cells as nano-surrogates for their origin cells; on the other hand, they may be released from cells and send biological messages to other cells. Exos are thought to be one subtype of the most thoroughly studied EVs, which form inward multivesicular bodies through the endolysosomal approach and are released from the fusion of multivesicular bodies with the plasma membrane [8]. Exos are secreted by most cell types in the central nervous system (CNS) and the peripheral system, including microglia, neurons, astrocytes, mesenchymal stem cells, plasma, cerebrospinal fluid, etc. [9] (Figure 1). They encapsulate a range of biomolecules including microRNA (miRNA), mRNA, and proteins, and participate in intercellular crosstalk under physiological and pathological conditions [6]. In detail, they interact with other cells on the cell surface through a specific receptor or by mixing the cargoes with the cellular contents after endocytosis [10]. Meanwhile, Exos are considered a new way of facilitating intercellular communication by delivering information through tissues to the target cells of numerous diseases, including neuroinflammatory diseases, neurodegenerative diseases, neoplastic diseases, and autoimmune diseases [11,12]. Moreover, Exos are practical for long-term storage and easily pass through the blood–brain barrier (BBB) while protecting their molecules wrapped in their bilayer lipid structure [13,14]. Indeed, recent studies have noted that Exos can be used as potential diagnostic markers in TBI and that these nanosized vesicles have a significant role in intracellular communication, serving as cargo for the intercellular delivery of biomaterials [15].

MiRNAs are a class of small and non-coding endogenous RNA molecules that regulate the expression of various genes at the post-transcriptional level via binding to complementary seed sequences in the 3′-untranslated regions of target mRNAs, resulting in mRNA degradation or translational repression [16]. Numerous studies have found that miRNAs play a critical role in the pathological processes of different diseases. Accumulating evidence has also shown that dysregulated miRNAs can be discovered in both in vitro and in vivo TBI models [17,18]. As mentioned previously, Exos may carry various molecular constituents of cellular origin, including miRNAs. 

The action of Exos has been extensively studied among EVs, especially Exos-derived miRNAs as cargoes (Figure 1).To date, there are no biological tools that detect mild TBI or monitor brain recovery. Given that TBI patients require new diagnostic approaches to identify neurotrauma and predict the risk of neurological injury, endogenous markers must be considered. Several miRNAs target key neuropathophysiological pathways associated with TBI [15,17]. Therefore, the aim of this review is to analyze Exos and their miRNA cargoes as treatment strategies after TBI, to provide new strategies for preventing the long-term progression of the disease, and to provide new literature for the diagnosis and treatment of TBI by cell-derived Exos.

## 2. Cell-Derived Exosomes and Exosome-Derived microRNAs in TBI

### 2.1. Microglia-Derived Exosomes and Exosome-Derived microRNAs in TBI

Microglia are resident mononuclear phagocytes that comprise 10–15% of all cells in the central nervous system (CNS) [19]. Numerous studies have confirmed that microglia play a key role in maintaining CNS homeostasis, axon outgrowth, synaptic plasticity, and neuronal death in physiological conditions [20,21,22]. Exosomes are endogenous micro-vesicles that play pivotal roles in intercellular signaling by transporting functional cargoes such as RNA, lipids, and proteins from one cell to another [23]. In the CNS, a comprehensive connection between neurons and glial cells exploits secreted exosomes responsible for long-distance intercellular communication [24]. Recent studies have shown that microglia-derived Exos (MD-Exos) play a critical role in microglia–neuron interaction in both healthy and pathological brains. MD-Exos exhibit a strong interaction with axons in physiological conditions and significantly increase the outgrowth of axons [25,26]. However, in neurodegenerative diseases such as Parkinson’s disease and Alzheimer’s disease, Exos derived from activated microglia deliver misfolded proteins to neurons, leading to neuronal dysfunction and death [27,28]. 

Furthermore, studies have revealed that microglia aggregate onto the damage site within minutes and undergo time-dependent and injury-related transcriptional profile changes after TBI [29]. In the acute stage following TBI, activated microglia exacerbate neuronal death and impede neurite outgrowth and synapse recovery [29,30]. It can be hypothesized that MD-Exos also affect CNS homeostasis after TBI.

Increasing studies have shown that MD-Exos can mediate microglia–neuron interactions after TBI. A recent study demonstrated that injured microglia partially suppress neurite outgrowth and synapse recovery by downregulating exosome-derived miR-5121 delivered to injured neurons after TBI. A severe stretch-induced injury increases Exos release in microglia, and neurons absorb these MD-Exos in vivo and in vitro. Stretch-injured MD-Exos decreased the neurite outgrowth and the dendritic complexity in stretch-injured neurons, reducing the neuron percentage and the apical dendritic spine density in the peri-contusion region after TBI. Meanwhile, the miR-5121 level decreased most significantly in stretch-injured MD-Exos. The overexpression of miR-5121 in stretch-injured MD-Exos partially reversed the inhibition of neurite outgrowth and neurons’ synaptic recovery by directly targeting RGMa after TBI. Moreover, the motor coordination in miR-5121 overexpressed MD-Exos treated mice was significantly improved after TBI [31]. One interesting study revealed that MD-Exos-derived miR-124-3p contributes to alleviating neurodegeneration and improving cognitive outcomes after TBI by transferring into neurons and targeting the Rela/ApoE signaling pathway. The levels of MD-Exos-derived miR-124-3p from the injured brains were significantly altered from the acute to chronic phases after TBI. Exos-derived miR-124-3p treatment alleviates the neurodegeneration of cultured neurons after injury by regulating neurodegenerative indicators’ expression, promoting neurite outgrowth, and inhibiting the abnormalities of Aβ. Additionally, the MD-Exos injected into TBI mice were absorbed by neurons in the injured brain [32]. It was also reported that MD-Exos enriched with miR-124-3p could attenuate trauma-induced, autophagy-mediated neuronal injury via targeting FIP200 to inhibit the activity of FIP200-mediated neuronal autophagy in vitro. Researchers added brain extracts harvested from TBI model mice to cultured BV2 microglia in vitro to imitate the microenvironment of TBI [33]. Another study indicated that MD-Exos-derived miR-124-3p suppresses neuronal inflammation and promotes neurite outgrowth via Exos-to-neurons transfer by targeting the PDE4B/mTOR signaling pathway, thus improving the neurologic outcome and inhibiting neuroinflammation in TBI mice [34].

EVs also propagate neuroinflammation from the CNS to the circulatory system. A study was designed to determine the role of microglia-derived microvesicles (MD-MVs) in TBI, and researchers found that the number of MVs increased within 24 h after injury. These MVs originated from microglia, suggesting that MD-MVs were released from the brain after TBI and then reached the circulatory system. Moreover, MD-MVs isolated from TBI were injected into un-injured mice, demonstrating that these MD-MVs loaded with pro-inflammatory molecules such as miR-155, TNF-a, and IL-1β could transfer the post-traumatic neuroinflammatory phenotype into animals. This evidence suggests that MVs can propagate neuroinflammation in TBI [7].

Therefore, miR-5121, miR-124-3p, and miR-155 from MD-Exos or MD-MVs can be novel therapeutic targets for interventions of neurodegeneration and neuronal inflammation after TBI. Furthermore, miRNA-manipulated MD-Exos may provide promising therapy for TBI (Table 1).

### 2.2. Astrocyte-Derived Exosomes and Exosome-Derived microRNAs in TBI

Astrocytes are the most abundant glial cells in the human brain, numbering five times that of neurons, and play a significant role in the brain in maintaining the BBB, regulating synaptic circuits, participating in neurotransmitter recycling, providing metabolic support for neural tissue, regulating regional blood flow, and repairing and scarring processes of the brain following various injuries [35,36,37,38,39]. The diversity and essence of astrocyte functions establish its predominance among other cells of the CNS.

Astrocyte-derived Exos (AD-Exos) and astrocyte-derived EVs (AD-EVs) are enriched with a variety of biological molecules including miRNAs. It has been reported that Exos secreted by astrocytes may be channels for transferring the cargo of macromolecular substances to nearby and distant cells, resulting in an extensive range of functional changes in recipient cells [40]. AD-EVs have also been proposed to confer neuroprotective functions by regulating neural uptake, differentiation, and firing [41,42]. Interestingly, AD-Exos and AD-EVs secreted in normal conditions are known to be enriched with neurotrophic and neuroprotective effects. On the contrary, Exos and EVs released by astrocytes in abnormal conditions such as nutrient deficiency, oxidative stress, and inflammation exert a neuroprotective effect and improve neurite regeneration and outgrowth [43,44,45]. TBI is a common brain disease. As cell-derived proteins, AD-Exos are considered at significantly higher levels than those Exos derived from neurons and other glial cells in CNS. These AD-Exos can be considered biomarkers for TBI [23]. Numerous studies have demonstrated that Exos are closely related to TBI [15,17,18], but it is unclear how AD-Exos protect brain functions after TBI.

Firstly, astrocytes and neurons regulate each other through Exos and EVs. A few interesting studies conducted on experimental TBI models reported that AD-Exos could repair and protect injured neurons. Exos were absorbed by neurons and released GJA1-20 k protein after TBI in a rat model. These AD-Exos secreted GJA1-20 k proteins reduced the phosphorylation of Cx43, protected and restored mitochondrial function, and down-regulated the apoptosis rate, thereby promoting neuronal functional recovery [46]. AD-Exos also reduced the water content and lesion volumes in both the cortex and hippocampus, reduced the neuronal cell loss and atrophy of the TBI in both rat and mouse models, protected against TBI-induced neuronal mitochondrial oxidative stress and apoptosis, and significantly attenuated TBI-induced short-term and long-term memory and learning deficits via the Nrf2/HO-1 signaling pathway [47]. In another mouse model study, the upregulation of NKILA in AD-EVs upregulates NLRX1 via suppressing miR-195, thereby exerting a suppressive effect on neuronal apoptosis and damage and promoting neuronal proliferation in TBI [48]. Furthermore, a clinical study demonstrated that AD-Exos levels were elevated by the increase in actors’ Bb and D of the alternative pathway and the complement component C4b of the classical pathway in acute sports-related TBI (sTBI) patients. The three complement pathways rapidly improved the increased AD-Exos levels of C5b-9 TCC and C3b, both of which affected injured synapses and damaged neurons. AD-Exos levels of the complement regulatory membrane proteins CD59 and CR1 declined a few days after sTBI. However, the return of AD-Exos levels of C5b-9 TCC and CR1 to those of controls required decades in patients after sTBI. Their results demonstrated that AD-Exos complements and other protein cargoes should be considered when evaluating biomarkers of sTBI [23]. Overall, these studies revealed and clarified the important functions of AD-Exos and AD-EVs in the astrocyte–neuron protection process after TBI.

Secondly, astrocytes and microglia regulate each other through Exos in TBI. One study explored how miRNAs secreted by Exos derived from activated astrocytes play a significant role in the astrocyte–microglia interaction. AD-Exos enriched with miR-873a-5p could promote the M2 phenotype transformation of microglia in the early stage of TBI and inhibit microglia-mediated neuroinflammation, thus improving neurological deficits after TBI by inhibiting the Erk/NF-κB signaling pathway. Meanwhile, the anti-neuroinflammatory effect of miR-873a-5 can also reduce brain edema in a mice TBI model. In addition, the expression of miR-873a-5p in the brain tissue of TBI patients was detected in this study [49].

Thirdly, apoptosis contributes to the pathogenesis of TBI [50], and AD-Exos might exert a therapeutic effect after TBI by suppressing apoptosis. Researchers incorporated plasmids expressing Bcl-2 and Bax shRNA into AD-Exos to overexpress the Bcl-2 gene and silence the Bax gene in recipient cells. The final data indicated that these modified AD-Exos could suppress apoptosis and ameliorate neurological and functional deficits in a mouse model of TBI. Moreover, these modified AD-Exos could also attenuate the impairment of long-term potentiation (LTP) and miniature excitatory postsynaptic currents (mEPSCs) in the hippocampus of TBI mice [51].

Neuroinflammation is one of the hallmarks of several neurodegenerative diseases and disorders, and AD-Exos have the potential to serve as neuro-inflammatory biomarkers in TBI. In an in vitro study, primary human astrocytes were activated by IL-1β stimulation, and AD-Exos-derived miRNA cargoes released in a neuroinflammatory stress model were examined. The results suggested that IL-1β-induced acute neuroinflammation and oxidative stress release a very distinct miRNA expression profile such as miR-141-3p or miR-30d through Exos, which may regulate the inflammatory response [52].

Finally, we noted that the plasma AD-Exos levels of Aβ42, postsynaptic protein, and NRGN could discriminate between military service personnel with mild traumatic brain injury (mTBI) and those without TBI with moderate sensitivity and accuracy. Injury-induced aggregation of Aβ42 and P-396-tau activates the phagocytic response in astrocytes; this response may lead to higher protein levels within astrocytes, which are subsequently expelled and trafficked to the periphery via plasma AD-Exos in mTBI patients. In addition, plasma AD-Exos were not toxic to neuron-like cells. Plasma AD-Exos contained detectable levels of Aβ42 and p-tau, and these cargo proteins were still not toxic to neuron-like cells in vitro [53]. (Table 2)

### 2.3. Neuron-Derived Exosomes and Exosome-Derived microRNAs in TBI

Neuron-derived exosomes (ND-Exos) collect neuronal proteins of cellular membranes and cytosol through the endosomal pathways prior to being secreted into the extracellular fluid of the CNS and entering into other neuronal cells or passing through the BBB into the blood [54]. ND-Exos and ND-Exos-miRNAs could also affect neurological recovery following TBI (Table 3).

In the CNS, neurons and microglia influence each other through Exos in TBI. A report by Yin et al. showed that ND-Exos containing miR-21-5p were phagocytosed by microglia and induced microglial polarization, while the miR-21-5p expression increased in M1 microglia. The polarization of M1 microglia accelerated the release of neuroinflammatory factors, increased the accumulation of P-tau protein, inhibited the neurite outgrowth, and promoted the apoptosis of neurons, which formed a model of cyclic cumulative damage similar to the TBI model. Therefore, regulating the expression of miR-21-5p or the secretion of ND-Exos may be a novel strategy for treating neuroinflammation in TBI [55].

Neurons improve neuronal autophagy through Exos in TBI. A study indicated Rab11a as a potential target gene of miR-21-5p, and miR-21-5p acts as cargo in the Exos playing a critical role in the inhibition of neuronal autophagy. The increased levels of miR-21-5p in ND-Exos can be translocated to damaged neurons to regulate excessive neuronal autophagy by inhibiting Rab11a, thereby resulting in protective effects against neuronal injury after TBI [56].

There is growing evidence that plasma ND-Exos cargoes have diagnostic and predictive value as biomarkers in patients after TBI. It has been suggested that the slope of change in ND-Exos synaptopodin (SYNPO) between 8 h and 14 h after TBI is a promising biomarker for three clinical study outcomes [57]. A clinical study by Goetzl et al. showed that relative to levels in those without cognitive impairment after TBI, the levels of plasma ND-Exos proteins, including claudin-5, annexin VII, and aquaporin 4, were increased in subjects with cognitive impairment after TBI for at least one year after a sports-related TBI. This group of ND-Exos proteins was increased relative to controls for at least one year after sports-related TBI and in older adults in veterans’ homes up to seven decades after TBI, including PrPc, synaptophysin-3, P-T181-tau, P-S396-tau, Aβ42, and IL-6 [58]. Another clinical study illustrated that altered levels of plasma ND-Exos and their cargo proteins characterize acute and chronic mild TBI [59]. Besides, as mentioned above, plasma ND-Exos levels of Aβ42, postsynaptic protein, and NRGN could discriminate between military service personnel with mTBI from those without TBI with moderate sensitivity and accuracy. Plasma ND-Exos were also toxic to recipient cells, containing detectable levels of p-tau and Aβ42. However, unlike plasma AD-Exos, the cargo proteins from these plasma ND-Exos in patients with mTBI were toxic to neuron-like cells in vitro [53].

### 2.4. Mesenchymal Stem Cell-Derived Exosomes and Exosome-Derived microRNAs in TBI

Mesenchymal stem cells (MSCs) are multipotent stromal cells obtained from various sources, including bone marrow, the umbilical cord, and adipose tissue [60,61]. MSCs show biological characteristics such as minimal immunogenicity, immune regulation, multi-differentiation potential, and culture expandability [62,63]. MSC-based therapies have shown promise in treating inflammatory diseases, including TBI [64,65,66,67]. Post-TBI MSCs therapy has recently received considerable attention, with various animal studies indicating neurological functional recovery after treatment and certain studies already being translated into clinical settings [68,69]. Nonetheless, as with most novel treatments, some limitations exist for their clinical application [70]. A major setback in MSC therapy is the reported tendency of potential tumorigenicity, with another being that only a small fraction of transplanted MSCs survive and differentiate into neurons [64]. Moreover, relatively small amounts of MSCs cause cerebral ischemia by intracerebral injection, the distribution of MSCs throughout the body by intravenous injection, and uncontrolled growth of the transplanted cells [71]. 

Our initial hypothesis was that MSCs could readily differentiate into neurons or other cells in the injured brain after transplantation. However, only a limited number of MSCs undergo such differentiation. The paracrine factors secreted by MSCs are now suggested to exert therapeutic effects rather than differentiating MSCs [72]. Abundant evidence has revealed the paracrine action of Exos/EVs and soluble factors secreted by MSCs as a promising strategy to address the aforementioned problems. Treatment with MSC-derived exosomes (MSCs-Exos) demonstrates a neuroprotective effect by improving neurobehavior performance, promoting neurogenesis and angiogenesis, reducing neuroinflammatory responses, and interfering with cells in CNS including astrocyte, neurons, and microglia [73,74,75,76].

Firstly, MSCs-Exos may play a vital role in functional improvement after TBI by promoting neuronal recovery function. In an interesting study involving a significantly high expression of miR-216a-5p, BDNF-induced bone-marrow-derived MSCs-Exos (BM-MSCs-Exos) improved neuronal regeneration and cell migration and inhibited inflammation and apoptosis in vivo and in vitro, thus promoting the recovery of spatial learning abilities and sensorimotor functions [77]. 

Then, MSCs-Exos may exert a positive neuroprotective effect by interfering with astrocytes after TBI. A study observed that human umbilical cord derived MSCs-Exos (UC-MSCs-Exos) could incorporate into hippocampal astrocytes and reduce reactive astrogliosis and inflammation in vitro and in vivo after TBI. UC-MSCs-Exos could also ameliorate LPS-induced calcium signaling dysregulation, mitochondrial dysfunction, and SE-induced learning and memory impairments in mice. The Nrf2-NF-κB signaling pathway was also involved in inhibiting astrocytic activation by UC-MSCs-Exos [78]. MSCs could also target microglia via Exos and EVs in TBI. Investigators identified that BM-MSCs-Exos secreted miR-32-3p, which targeted DAB2IP, inducing microglia autophagy without affecting the proliferation and growth of microglia in TBI. In this study, miR-211-3p, miR-188-5p, and miR-465-5p may also have been involved in microglial autophagy; however, the researchers did not study them in depth [79]. A recent study demonstrated that BM-MSCs-Exos reduced the lesion size, inhibited the expression of Bax and TNF-α, and enhanced the expression of Bcl-2, thereby serving a neuroprotective function by ameliorating early neuroinflammatory responses in TBI mice via modulating the polarization of microglia/macrophages [80]. A paper by Zhang et al. showed that miR-124 enriched with BM-MSCs-Exos promoted the M2 polarization of microglia and hippocampal neurogenesis by inhibiting the TLR4 pathway, resulting in functional recovery after TBI [81]. In addition, scholars suggested that intracerebroventricularly microinjected human adipose-derived MSCs-Exos (AD-MSCs-Exos) specifically entered microglia/macrophages and suppressed their activation by inhibiting the NF-κB and MAPK signaling pathway after TBI, thereby increasing neurogenesis, suppressing neuroinflammation, reducing neuronal apoptosis, and promoting functional recovery [82]. These findings confirm that MSC-Exos can reverse the inflammation following TBI, reduce the progression of secondary injury and prevent further neuronal cell loss via regulating the two key resident immune cells—microglia and astrocytes.

Moreover, a few studies validated that MSCs-Exos ameliorate BBB integrity after TBI. In a series of studies on the TBI swine model by Williams et al., early treatment with a single dose of BM-MSC-Exos significantly attenuated cerebral swelling and lesion size with a corresponding decrease in intracerebral pressure; decreased levels of blood-based cerebral biomarkers including GFAP, laminin, and claudin-5; decreased albumin extravasation; and improved BBB integrity [83]. Subsequent RNA sequencing data analysis showed that the BM-MSC-Exos treatment significantly increased gene expressions associated with neuronal development, synaptogenesis, neurogenesis, and neuroplasticity. The treatment also significantly reduced gene expressions associated with neuroinflammation, neuroepithelial cell proliferation, stroke, and nonneuronal cell proliferation contributing to reactive gliosis and increasing the gene expressions associated with the stability of BBB [84]. The levels of inflammatory markers (interleukin IL-1, IL-6, IL-8, IL-18, and NF-κB) and BAX were down-regulated, whereas the levels of granulocyte-macrophage colony-stimulating factor and brain-derived neurotrophic factor were up-regulated through BM-MSC-Exos treatment [85]. Additionally, the role of BM-MSCs-Exos in improving neurological functions after TBI was also confirmed by Williams et al. in another article [86].

Lastly, MSCs-Exos have shown promise in treating brain disorders, with some results highlighting neuroprotective effects through anti-neuroinflammation, pro-neurogenesis, and pro-angiogenesis after TBI. A certain research group developed an in vivo assay for BM-MSCs-EVs efficacy in suppressing neuroinflammation after TBI in mice and found that intravenous infusion of those isolated BM-MSCs-EVs shortly after inducing TBI improved spatial learning and pattern separation deficits 1 month later [75]. Another research group reported that BM-MSCs-Exos significantly improved cognitive and sensorimotor functional recovery, increased the number of newborn neuroblasts and mature neurons in the hippocampal dentate gyrus, increased the number of newborn endothelial cells in the lesion boundary zone and hippocampus, and reduced brain inflammation. In conclusion, the study demonstrated that intravenous administration of BM-MSCs-Exos may improve functional recovery and promote neurovascular remodeling including angiogenesis and neurogenesis and reduce neuroinflammation in rats after TBI [73]. In addition, BM-MSCs-Exos cultured with collagen scaffolds could also achieve similar effects in improving functional recovery of TBI rats [74]. A further study confirmed a wide range of effective doses for the treatment of TBI with BM-MSCs-Exos therapy for at least 7 days of the therapeutic window post-injury [87]. Moreover, they have verified that miR-17-92 cluster-enriched BM-MSCs-Exos significantly improved histological and functional recovery after TBI [88].

In conclusion, the protein components and miRNA of MSCs-Exos are involved in stimulating various biological processes in the target cells through neurite regrowth and neuron survival, microglia polarization, exosome biogenesis, cellular motility, BBB integrity, immunomodulation, inflammation regulation, neurogenesis, angiogenesis, extracellular matrix modification, antioxidation, self-renewal, or differentiation (Table 4).

### 2.5. Other Stem Cell-Derived Exosomes and Exosome-Derived microRNAs in TBI

Despite various MSC-related studies, some scientists still turn their attention to other stem cells Exos (SC-Exos) or stem cell EVs (SC-EVs) for the treatment of TBI (Table 5).

Adipose-derived stem cells (ADSC-Exos) might have beneficial biological functions after TBI. On the one hand, Exos derived from human ADSC-Exos reduce motor and cognitive impairments following TBI. ADSC-Exos containing MALAT1 may reduce cortical damage via modulating the NRTK3 (TrkC)/MAPK pathway, and the therapeutic window would span at least 48 h [89]. On the other hand, long noncoding RNA MALAT1 in ADSC-Exos drove regenerative function and modulated inflammation-linked networks following TBI. Researchers reported that a brain and spleen transcriptome analysis using RNA Seq showed a MALAT1-dependent modulation of regeneration, inflammation, cell cycle, and cell-death-related pathways. Importantly, MALAT1 regulated the expression of other non-coding RNAs (ncRNA) including snoRNAs [90]. In summary, ADSC-Exos have a beneficial role in modulating pathology after TBI, particularly those containing MALAT1.

Similar to MSCs, SC-Exos reverse inflammation following TBI via regulating the key resident immune cells and neurogenesis in the CNS. SC-Exos derived from human exfoliated deciduous teeth (hEDSC-Exos) may promote functional motor recovery and decelerate neuroinflammation by shifting microglia M1/M2 polarization in the TBI rats model [76]. Human embryonic neural stem cell-derived extracellular vesicle (heNSCs-EVs) treated rats presented significantly decreased lesion sizes, an increased presence of endogenous NSCs, and facilitated motor function recovery 4 weeks after TBI. Simultaneously, a positive correlation between VEGFR2 and Nestin expression around the injury sites in heNSCs-EVs-treated rats was observed, potentially suggesting a VEGFR2-dependent increase in NSC proliferation and recruitment after TBI [91].

### 2.6. Humoral Cell-Derived Exosomes and Exosome-Derived microRNAs in TBI

Although the cause of TBI is largely unavoidable, the consequently impaired responses including inflammatory response to injury may require therapeutic intervention [92]. However, targeting the damaged site within the brain is problematic owing to the presence of the BBB. Exos cross the BBB and are detected in the peripheral circulation. The BBB is uniquely positioned to signal CNS injury to the systemic immune system and, as a consequence, recruit immune cells to the parenchyma. There are two main pathways that mediate this communication: neural and humoral. Due to the characteristics of easy penetration of the BBB, humoral cells, including peripheral blood cells (PBC) and cerebrospinal fluid (CSF) cells, as well as humoral cell-derived exosomes (HCD-Exos) and biomarker-related studies, have aroused the personnel’s interest (Table 6). However, it remains unclear whether these peripherally accessed Exos and biomarkers reflect CNS processes [93].

Since peripheral blood is easier to collect than damaged brain tissue, a large number of HCD-Exos-related clinical studies have been carried out. The discovery of possible trauma-specific biomarkers in peripheral blood samples could serve as a de-facto “liquid biopsy” for TBI and could greatly aid clinicians in making the correct differential diagnosis and assessment of TBI. A multicenter study of 195 veterans involved in the chronic effects of neurotrauma consortium suggested that repetitive chronic TBI may contribute to chronic neuropsychological symptoms including loss of consciousness or posttraumatic amnesia. Outcomes in TBI patients were associated with elevated p-tau and tau from plasma cell-derived exosomes (PCD-Exos) that may provide peripheral sources of central and informative biomarkers in remote TBI [94]. Similarly, in a study of military personnel with mild TBI, investigators compared the level of amyloid-beta (Aβ42), tau and IL-10 between volunteers with mild TBI and those without in PCD-Exos-enriched neuronal origins, the concentrations of the above three indicators were increased. These tau elevations were associated with chronic post-concussive symptoms, while higher IL-10 levels were associated with symptoms of post-traumatic stress disorders (PTSD) [95]. In another study of older veterans with TBI, increased levels of blood-based, CNS-enriched PCD-Exos biomarkers—including p-tau, NFL, IL-6, TNF-α, and GFAP—and cognitive impairment could be detected even decades after TBI [96]. In a clinical longitudinal study of moderate-to-severe TBI, concentrations of NFL and GFAP from PCD-Exos were significantly higher in patients with diffuse injury than those with focal lesions. The concentration of ubiquitin carboxy-terminal hydrolase L1 (UCH-L1) in the PCD-Exos profile, characterized by a sharply increasing value and a secondary steep rise, was related to early mortality with a sensitivity and specificity of 100% [97]. In another study, PCD-Exos proteins in plasma from TBI patients stratified by the Glasgow Coma Score (GCS) was investigated to identify specific markers of injury severity [98]. Additionally, Puffer et al. showed that GFAP expression in plasma cell-derived EVs (PCD-EVs) was approximately 10-fold higher in TBI patients with altered consciousness than in patients with normal consciousness, and identified 11 highly differentially expressed PCD-EVs-miRNAs targeting the biologically relevant cellular pathways, including organismal injury, cellular development, and organismal development [99].

The levels of miRNAs in Exos and EVs of peripherally circulating cells have been assessed in several clinical TBI-related studies in search of promising biomarkers. Vorn et al. profiled peripheral blood cell-derived Exos (PBCD-Exos) and their microRNAs from young adults with or without chronic mild TBI and identified 25 significantly dysregulated microRNAs associated with pathways of neurological disease, psychological disease and organismal injury and abnormalities, thereby resulting in the possibility of long-lasting post-injury symptoms. Moreover, pathway analysis revealed that 14 PBCD-Exos-miRNAs were associated with neurological disorders, 13 with psychological disorders and 23 with organismal damage and abnormalities [100]. Devoto et al. extracted and analyzed PCD-Exos-miRNAs from 153 mild TBI military personnel and showed that 17 PCD-Exos-miRNAs were dysregulated in the entire mild TBI group and 32 were dysregulated in the repetitive mild TBI group, which correlated with pathways of cell development, neurological disease and inflammatory and neuronal repair. TBI history and neurobehavioral symptom survey scores were negatively and significantly correlated with miR-103a-3p expression [101]. One study in military personnel with blast-related chronic mild TBI confirmed that 32 PCD-EVs-miRNAs and 45 PCD-EVs-miRNAs targeting neuronal function, vascular remodeling, BBB integrity, and neuroinflammation pathways were significantly changed [102]. Another study of chronic PTSD symptoms in service members and veterans with mild TBI demonstrated that in participants with more severe PTSD symptoms and positive mild TBI history, NFL and miR-139-5p levels of PCD-EVs were increased. Meanwhile, other PCD-EVs-miRNAs were implicated in pathways of glucocorticoid receptor, neurodegeneration, and inflammation [103]. The PCD-Exos-miRNAs analysis of the TBI rat model studies showed that there were 50 significantly differentially expressed miRNAs, and the most relevant pathways were the MAPK, Rap1, Ras, and regulation of actin cytoskeleton-related signaling pathways [104]. Ginsenoside Rg1 decreased the level of PCD-Exos-miR-21, inhibited both PCD-Exos and miR-21 from peripheral blood flow towards the brain, increased the expression of TJPs (ZO-1, occluding, and claudin-5) and MMP, and elevated the expression of GFAP through NF-κB signaling pathway after TBI in rats, thereby improving cerebral vascular endothelial injury and BBB integrity [105]. Ko et al. isolated brain cell-derived GluR2+ EVs directly from plasma (PBCD-EVs) in a TBI mice model and clinical patients; PBCD-EVs-miRNAs and associated signaling pathways were analyzed. They generated a panel of four miRNAs (miR-203b-5p, miR-203a-3p, miR-206, miR-185-5p) for TBI patients and eight miRNAs (miR-150-5p, miR-669c-5p, miR-488-3p, miR-22-5p, miR-9-5p, miR-6236, miR-219a.2-3p, miR-351-3p) for TBI mice [106,107].

Similar to AD-Exos, ND-Exos, and MSCs-Exos, recently published studies revealed that endothelial cell-derived Exos/EVs/MVs may also promote angiogenesis and BBB continuity after TBI. It is well known that brain microvascular endothelial cells (BMECs) are one of the major components of BBB that maintain brain homeostasis. Endothelial progenitor cell-derived microvesicles have been reported to promote angiogenesis in rat BMECs in vitro [113]. In an experimental mice model, plasma endothelial cell-derived microvesicles (PECD-MVs) contained elevated levels of occludin, leading to vascular remodeling of BBB [108]. Human endothelial colony-forming cell-derived exosomes from umbilical cord blood (hbECFCD-Exos) have been verified to increase occludin and ZO-1 expression and restore BBB continuity in TBI mice by targeting the PTEN/AKT signaling pathway. Delivery of these hbECFCD-Exos to TBI mice exerted significant protective effects, which were associated with reduced MMP-9 expression, less EB dye extravasation, and tight junction protein degradation [109].

Furthermore, although CSF collection is more difficult than peripheral blood, few investigators have recently designed CSF cell-derived Exos/EVs/MVs (CSFCD- Exos/EVs/MVs) related studies in TBI patients. Manek et al. have found elevated levels of GFAP, UCH-L1, presynaptic terminal protein synaptophysin and αII-spectrin breakdown products (BDPs) in CSFCD-MVs/Exos from severe TBI patients. These CSFCD-MVs/Exos were likely derived from various brain cell types after TBI—neurons and astroglia, for example [110]. The researchers examined the protein composition of CSFCD-EVs in players with cognitive and neuropsychiatric dysfunction, as well as a history of contact sports or TBI. The levels of t-tau and p-tau_181_ in CSFCD-EVs were positively correlated with the levels of t-tau and p-tau_181_ in total CSF in those participants, respectively. MAPT (tau) may be a potential upstream regulator for predicting the risk of chronic traumatic encephalopathy in TBI players [111]. It should be noted, however, that it was a small sample study. Furthermore, severe TBI has been reported to induce early changes in the protein composition and physical properties of CSFCD-EVs associated with nerve regeneration within 7 days of injury [112].

### 2.7. Other Cell-Derived Exosomes and Exosome-Derived microRNAs in TBI

Purely humoral pathways were believed to be responsible for initiating the acute phase response after TBI; however, new studies have raised doubts about these views. Studies have revealed that vagotomized animals continue to present a peripheral response to CNS injury [114]. Although the relationship between local and systemic production of cytokines/chemokines after the injury is a key determinant of local leukocyte recruitment after CNS injury [115], no cytokine/chemokine pathways were identified from the CNS to the periphery. Evidence suggests that the amount of EVs from macrophage/monocyte populations (MPD-EVs), brain vascular endothelial cells (BVEC-EVs), and plasma (PD-EVs) rapidly increased after TBI, accompanied by an increase in CNS and hepatic leukocyte recruitment. Meanwhile, peripheral inflammation occurs via the manipulation of the circulating EVs population with “primed” EVs, exacerbating the injury to the CNS [116].

Saliva, intriguingly, also secretes EVs [117,118]. A clinical study enrolled 54 patients including saliva-derived EVs (SD-EVs) sampling, of which 16 patients enrolled from an outpatient concussion clinic and 15 from the emergency department who sustained brain trauma within 24 h. Many Alzheimer’s disease-related genes were assayed, and the results showed that certain mRNA as well as CDC2, CSNK1A1, and CTSD may also play a role in improving patients’ neuronal injuries in patients with TBI [119]. Meanwhile, several human inflammation-related genes from SD-EVs were also assayed in mixed martial artists with or without TBI. Several patterns of key gene expression and pathways acutely and chronically affected them after a head injury. Furthermore, a correlation was observed between absolute gene information signals and fight-related markers of the brain injury’s severity [120].

Furthermore, in an orthopedic-related study, investigators observed that miRNA-1224 was upregulated in bone marrow-derived extracellular vesicle (BM-EVs) cargoes of TBI mice. These BM-EVs activated NF-κB signaling genes and enhanced their colony-forming ability and osteoclast differentiation efficacy, leading to the reversal of bone loss and reduction in fracture rate [121] (Table 7).

### 2.8. Brain-Derived Exosomes and Exosome-Derived microRNAs in TBI

Despite the attraction of specific cerebral cell-derived Exos, few studies have focused on purely brain-derived Exos, EVs, or MVs (BD-Exos/EVs/MVs), regardless of the cell types (Table 8). TBI increases levels of miR-21, miR-146, and miR-7a and decreases the level of miR-212 in BD-EVs, while miR-21 shows the most changes. The BD-EVs-miR-21 expression was primarily localized to neurons near the lesion site of TBI, suggesting that miR-21 secretes from neurons as potential BD-EVs cargo [122]. Meanwhile, TBI-activated p-Cx43, which requires the activation of the ERK signaling pathway, mediates a nociceptive effect by propagating brain injuries and a neuroprotective effect by promoting hippocampal BD-EVs release [123]. We noted that BD-EVs-circRNAs and the circRNAs-miRNAs network might change after TBI, targeting the growth and repair of neurons signaling pathways, the development of the nervous system signaling pathways, and the transmission of nerve signaling pathways [124]. One study attempted to find potential monitoring biomarker candidate molecules in BD-EVs of gray matter tissues from the frontal cortex of deceased chronic traumatic encephalopathy (CTE) patients after TBI, and identified the cell type-specific molecules including p-tau, PLXNA4, SNAP-25, and UBA1 [125]. Another study also demonstrated that both tau and p-tau in BD-Exos after TBI were significantly elevated, and BD-Exos exacerbated motor and cognitive impairments [126]. On the contrary, the inhibition of BD-EVs release alleviated cognitive impairment after repetitive mild TBI [127].

Surprisingly, a research team working on coagulopathy after TBI revealed that lactadherin might prevent coagulopathy and improve the survival of severe TBI mice via promoting BD-MVs clearance [128]. Moreover, coagulopathy after TBI may also be treated by anticoagulation-targeting membrane-bound anionic phospholipids of BD-EVs to improve TBI outcomes [129].

## 3. Challenges and Opportunities

One of the major problems with the clinical application of cell-derived Exos or EVs treatment is the large variability in cell quality, due to the usage of different donors and their tissues, known as donor heterogeneity [130,131,132]. Developing production methods that minimize donor-to-donor and batch-to-batch variations requires robust quality control for the production lot of Exos. Other problems include short half-life, insufficient payload, rapid clearance after application, and limited targeting, resulting in limited clinical application. Meanwhile, the blood levels of Exos decreased rapidly after systemic application in both clinical and pre-clinical studies, and the circulation time of Exos was shortened by macrophage/microglial clearance. However, it is surprising that certain delivery methods, such as intranasal delivery of Exos, have shown unique advantages for easy use and its ability to cross the BBB efficiently, uptake by injury site, and lower dose requirement. In addition, despite their significant neuroprotective properties, limited research has been done on the potential of Exos-derived miRNAs in developing exosome-based therapies. There remains a lack of knowledge of the clinical implications of Exos-derived miRNA treatments.

Despite recent efforts to understand the role of various cell-derived EVs and Exos in CNS diseases, future research is required to determine the role of cell-derived Exos, especially their associated biomolecules, their presence in both healthy and abnormal cells, and their functional effect. Identifying the effect of cell-derived exosomes on both shorter and longer distance targets and their survival strategies may lead to the discovery and development of new diagnostic and therapeutic approaches. To further elucidate the roles of Exos/EVs in the pathophysiology of various CNS diseases, further narrative studies on various biomolecules in cell-derived Exos or cell-derived EVs are required.

Exos-based cell-free therapy delivers targeted regulatory genes such as miRNAs to enhance multifaceted aspects of neuroplasticity, reduce neuroinflammation, and enhance neurological recovery for effective and reparative therapies for several nerve injuries following TBI. In addition to the role of the miRNAs, further investigations of Exo proteins could fully investigate the active trophic mechanisms underlying Exos-induced therapeutic effects in TBI. Although Exos provide promising therapeutic effects in rodent models of TBI and some clinical patients, further investigations are required for clinical translation. Meanwhile, despite the critical and prominent role of Exos/EVs-derived miRNA in ameliorating TBI outcomes and modulating brain recovery using experimental models and clinical studies, further clinical studies are required to investigate human brain efficacy, limit side effects, and improve targeted mechanisms in the human body.

## 4. Conclusions

Overall, we show that some neural cells, especially neural stem cells, astrocytes and microglia, and their miRNAs such as miR-124, miR-141, miR-32, etc., may become promising strategies for targeted drug delivery in the treatment of TBI. However, in this review, we cover the broad topic of Exos and their miRNAs with TBI, and therefore it appears narrow to discuss Exos and their miRNA-related signaling pathways and molecular mechanisms in TBI. A more detailed summary is required in the future.

## Figures and Tables

**Figure 1 jcm-11-03223-f001:**
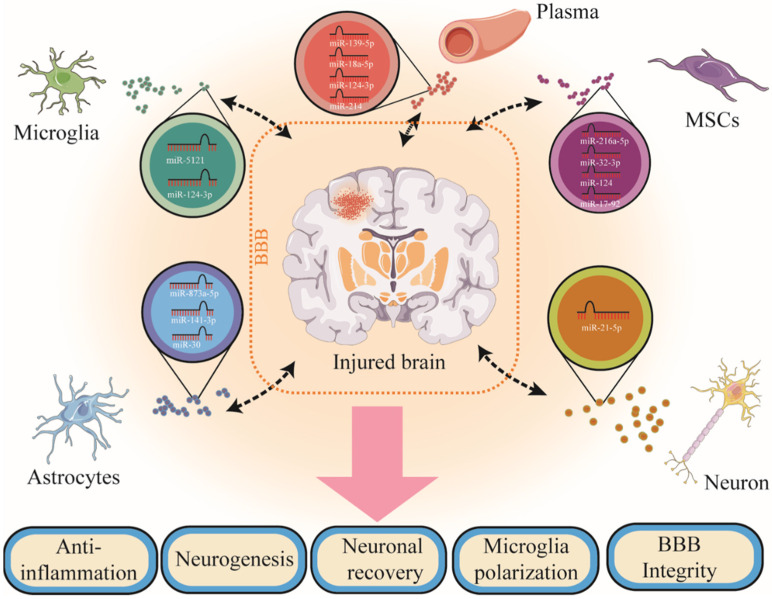
Exosomes are secreted by most cell types not only in CNS but also in the peripheral system, including microglia, astrocytes, neurons, mesenchymal stem cells (MSCs), and peripheral blood cells. The action of exosomes has been extensively studied, especially exosome-derived miRNAs as cargoes. Exosomes and exosome-miRNAs can easily pass through the BBB and affect neurological recovery following TBI.

**Table 1 jcm-11-03223-t001:** Microglia-derived Exos/MVs and MD-Exos/MVs-miRNAs studies in TBI.

Origin	Model Species	Involved miRNAs	Other Molecules	Biological Functions/Findings	Study
MD-Exos	mouse	miR-5121	RGMa	Promote neurite outgrowth and synapse recovery of neurons; improve motor coordination function	Zhao et al., 2021 [31]
MD-Exos	mouse	miR-124-3p	Rela ApoE	Promote neurite outgrowth of neurons; regulate neurodegenerative indicators expression; inhibit Aβ abnormalities; improve cognitive outcome	Ge et al., 2020 [32]
MD-Exos	mouse	miR-124-3p	FIP200	Inhibit neuronal autophagy; reduce neuronal injury	Li et al., 2019 [33]
MD-Exos	mouse	miR-124-3p	PDE4B mTOR	Suppress neuronal inflammation; promote neurite outgrowth	Huang et al., 2018 [34]
MD-MVs	mouse	miR-155	--	Propagate neuroinflammation from the CNS to the circulatory system	Kumar et al., 2017 [7]

MD-Exos, microglia-derived exosomes; MD-MVs, microglia-derived microvesicles.

**Table 2 jcm-11-03223-t002:** Astrocyte-derived Exos/EVs and AD-Exos/EVs-miRNAs studies in TBI.

Origin	Model Species	Involved miRNAs	Other Molecules	Biological Functions/Findings	Study
AD-Exos	rat	--	GJA1-20 k Cx43	Protect and repair damaged neurons; protect and restore mitochondrial function; down-regulate apoptosis rate	Chen et al., 2020 [46]
AD-Exos	rat mouse	--	Nrf2 HO-1	Reduce neuronal cell loss and atrophy; protect neuronal oxidative stress and apoptosis; attenuate memory and learning deficits	Zhang et al., 2021 [47]
AD-EVs	mouse	--	NLRX1 NKILA	Suppress neuronal injury and neuronal apoptosis; promote neuronal proliferation; enhance brain recovery	He et al., 2021 [48]
Plasma AD-Exos	human	--	complement C5b-9 TCC C3b, CR1	Repair injured synapses and damaged neurons; predict the prognosis of patients	Goetzl et al., 2020 [23]
AD-Exos	mouse human	miR-873a-5p	NF-κB Erk	Inhibit microglia-mediated neuroinflammation via microglia phenotype modulation; improve neurological deficits	Long et al., 2020 [49]
Modified AD-Exos	mouse	--	Bcl-2 Bax	Suppress apoptosis; ameliorate neurological and functional deficits	Wang et al., 2019 [51]
AD-Exos	mouse	miR-141-3p miR-30, et al.	--	Inhibit neuroinflammation and oxidative stress	Gayen et al., 2020 [52]
Plasma AD-Exos	human	--	Aβ42, p-tau NRGN postsynaptic protein	Discriminate military service personnel with mTBI from those without TBI	Winston et al. 2019 [53]

AD-Exos, astrocyte-derived exosomes; AD-EVs, astrocyte-derived extracellular vesicles; mTBI, mild TBI.

**Table 3 jcm-11-03223-t003:** Neuron-derived-Exos/EVs and ND-Exos/EVs-miRNAs studies in TBI.

Origin	Model Species	Involved miRNAs	Other Molecules	Biological Functions/Findings	Study
ND-Exos	mouse	miR-21-5p	--	Increase M1 microglia polarization; accelerate neuroinflammation factors release; increase the accumulation of p-tau protein; inhibit the neurite outgrowth; promote the apoptosis of neurons	Yin et al., 2020 [55]
ND-Exos	mouse	miR-21-5p	Rab11a	Regulate excessive neuronal autophagy; improve neuronal injury	Li et al., 2019 [56]
Plasma ND-Exos	human	--	SYNPO	Find a promising biomarker in TBI	Goetzl et al., 2018 [57]
Plasma ND-Exos	human	--	claudin-5 annexin VII aquaporin 4 PrPc, p-tau Aβ42, IL-6	Find promising biomarkers after remote TBI to improve cognitive impairment	Goetzl et al., 2020 [58]
Plasma ND-Exos	human	--	annexin VII claudin aquaporin 4 p-tau, et al.	Find promising biomarkers to characterize acute and chronic mTBI	Goetzl et al., 2019 [59]
Plasma ND-Exos	human	--	Aβ42 p-tau NRGN Postsynaptic protein	Discriminate military service personnel with mTBI from those without TBI	Winston et al., 2019 [53]

ND-Exos, neuron-derived exosomes; ND-EVs, neuron-derived extracellular vesicles; mTBI, mild TBI.

**Table 4 jcm-11-03223-t004:** Mesenchymal stem cell derived Exos/EVs and MSCs-Exos/EVs-miRNAs studies in TBI.

Origin	Model Species	Involved miRNAs	Other Molecules	Biological Functions/Findings	Study
BDNF-rBM- MSCs-Exos	rat	miR-216a-5p	--	Improve neuronal regeneration, cell migration; inhibit inflammation and apoptosis; improve spatial learning ability and sensorimotor function	Xu et al., 2020 [77]
hUC- MSCs-Exos	mouse	--	Nrf2 NF-κB	Improve inflammatory astrocyte alterations; improve mitochondrial dysfunction; improve learning and memory impairments	Xian et al., 2019 [78]
mBM-MSCs-Exos	mouse	miR-32-3p	DAB2IP	Induce microglia autophagy	Yuan et al., 2020 [79]
mBM-MSCs-Exos	mouse	--	Bcl-2 Bax TNF-α	Inhibit early neuroinflammation; modulate microglia/macrophages polarization; reduce the lesion size	Ni et al., 2019 [80]
rBM- MSCs-Exos	rat	miR-124	TLR4 NF-κB	Promote M2 polarization of microglia; promote hippocampal neurogenesis; promote functional recovery	Yang et al., 2019 [81]
hAD- MSCs-Exos	rat	--	NF-κB MAPK	Suppress microglia/macrophages activation; increase neurogenesis; suppress neuroinflammation; reduce neuronal apoptosis	Chen et al. 2020 [82]
hBM- MSCs-Exos	swine	--	GFAP laminin claudin-5	Attenuate cerebral swelling and lesion size; decrease blood-based cerebral biomarker level; improve BBB integrity	Williams et al. 2020 [83]
hBM- MSCs-Exos	swine	--	BDNF NTRK2 lipocalin 2 HIF1α et al.	RNA sequencing data analysis: improve neuronal development, synaptogenesis, neurogenesis, neuroplasticity, neuroinflammation, and stability of BBB	Williams et al. 2020 [84]
hBM- MSCs-Exos	swine	--	Interleukin NF-κB BAX et al.	Decrease inflammatory markers; decrease apoptotic markers; increase neurotrophic factor and granulocyte-macrophage colony-stimulating factor	Williams et al. 2020 [85]
hBM-MSCs -Exos	swine	--	--	Improve neurological functions	Williams et al. 2020 [86]
hBM- MSCs-EVs	mouse	--	--	Suppress neuroinflammation; improve spatial learning and pattern separation deficits	Kim et al., 2016 [75]
rBM- MSCs-Exos	rat	--	--	Promote neurovascular remodeling; reduce neuroinflammation; improve functional recovery	Zhang et al., 2015 [73]
hBM- MSCs-Exos	rat	--	--	2D or 3D cultured hUC-MSCs-Exos could promote neurovascular remodeling; reduce neuroinflammation; improve functional recovery	Zhang et al., 2020 [74]
hBM- MSCs-Exos	rat	--	--	Explore the range of effective doses and therapeutic window	Zhang et al., 2020 [87]
hBM- MSCs-Exos	rat	miR-17-92	--	Promote neurovascular remodeling; reduce neuroinflammation; improve functional recovery	Zhang et al., 2021 [88]

MSCs-Exos, mesenchymal stem cell derived exosomes; MSCs-EVs, mesenchymal stem cell derived extracellular vesicles; mTBI, mild TBI; rBM-, rat bone marrow derived; mBM-, mouse bone marrow derived; hBM-, human bone marrow derived; hUC-, human umbilical cord derived; hAD, human adipose derived.

**Table 5 jcm-11-03223-t005:** Other stem cell-derived Exos/EVs and SC-Exos/EVs-miRNAs studies in TBI.

Origin	Model Species	Involved miRNAs	Other Molecules	Biological Functions/Findings	Study
hADSC- Exos	mouse	MALAT1 (a lncRNA)	NRTK3 (TrkC) MAPK	Reduce motor and cognitive impairments; reduce the cortical damage	Moss et al., 2021 [89]
hADSC- Exos	rat	MALAT1 (a lncRNA)	--	Drive regenerative function; modulate inflammation-linked networks; MALAT1 affects mRNA and ncRNA expression	Patel et al., 2018 [90]
hADSC- Exos	rat	--	TNF-α IL-6	Shift microglia M1/M2 polarization; promote motor functional recovery; decelerate neuroinflammation	Li et al., 2017 [76]
heNSC -EVs	rat	--	VEGF VEGFR2	Increase endogenous NSCs and their migration; increase VEGF activity; promote recovery of motor function	Sun et al., 2020 [91]

SC-Exos, stem cell-derived exosomes; SC-EVs, stem cell-derived extracellular vesicles; hADSC-Exos, human adipose-derived SC-Exos; hEDSC-Exos, human exfoliated deciduous teeth-derived SC-Exos; heNSC-EVs, human embryonic neural stem cell-derived SC-EVs.

**Table 6 jcm-11-03223-t006:** Humoral cell-derived Exos/EVs and HCD-Exos/EVs-miRNAs studies in TBI.

Origin	Model Species	Involved miRNAs	Other Molecules	Biological Functions/Findings	Study
PCD-Exos	human	--	p-tau tau	Associate with the loss of consciousness or post-traumatic amnesia; find peripheral-to-central biomarkers	Kenney et al., 2018 [94]
PCD-Exos	human	--	Aβ42 tau IL-10	Identify biomarkers in plasma and PD-Exos that relate to chronic post-concussive and behavioral symptoms following TBI	Gill et al., 2018 [95]
PCD-Exos	human	--	p-tau TNF-α NFL IL-6	Determine whether blood-based biomarkers can differentiate older veterans with and without TBI and cognitive impairment	Peltz et al., 2020 [96]
PCD-Exos	human	--	NFL GFAP UCH-L1	Identify biomarkers in plasma and PD-Exos; PD-Exos NFL/UCH-L1 are sensitive indicators of axonal injury/early mortality, respectively	Mondello et al., 2020 [97]
PCD-Exos	human	--	many	Analyze differential protein expression in PD-Exos samples by mass spectrometry	Moyron et al., 2017 [98]
PCD-EVs	human	miR-1-3p miR-143-3p miR-151, et al.	GFAP et al.	Find promising biomarkers and pathways targeting consciousness	Puffer et al., 2020 [99]
PBCD-Exos	human	miR-223-3p miR-29b-3p miR-107, et al.	--	Find promising biomarkers in chronic mild TBI	Vorn et al., 2021 [100]
PCD-Exos	human	miR-139-5p miR-18a-5p et al.	TP53 IGF-1 TGF-β, et al.	Find promising biomarkers and pathways associated with pathobiology of chronic symptoms	Devoto et al., 2020 [101]
PCD-EVs	human	miR-106a-5p miR-106b-5p et al.	MME et al.	Identify biomarkers and pathways for blast-related chronic mild TBI	Ghai et al., 2020 [102]
PCD-EVs	human	miR-139-5p miR-3190-3p et al.	NFL Aβ-42 et al.	Find links between NFL and severity of PTSD symptoms; find links between persistent PTSD symptoms and PD-EVs-miRNAs levels	Guedes et al., 2021 [103]
PCD-Exos	rat	miR-106b-5p miR-124-3p et al.	MAPK Rap1 Ras, et al.	Find promising biomarkers and pathways	Wang et al., 2020 [104]
PCD-Exos	rat	miR-21 miR-21-5p	Rg1 GFAP NF-κB ZO-1, et al.	Improve cerebrovascular endothelial injury; protect the BBB integrity; restore neural function	Zhai et al., 2021 [105]
PBCD-EVs	mouse human	miR-203b-5p miR-203a-3p miR-206, et al.	MAPK PI3K-Akt et al.	Find promising biomarkers and pathways associated with TBI diagnosis	Ko et al., 2019 [106] 2020 [107]
hbECFCD- Exos	human	--	PTEN occludin AKT, ZO-1	Restore the BBB continuity; Reduce brain edema	Gao et al., 2018 [108]
PECD-MVs	mouse	--	occludin et al.	Improve vascular remodeling; restore the BBB continuity	Andrews et al., 2016 [109]
CSFCD-Exos/MVs	human	--	UCH-L1 GFAP, et al.	Find unique protein contents in CSF-Exos/MVs from severe TBI patients	Manek et al., 2018 [110]
CSFCD-EVs	human	--	MAPT p-tau_181_ t-tau, et al.	Find potential monitoring biomarkers in TBI players at risk for chronic traumatic encephalopathy	Muraoka et al., 2019 [111]
CSFCD-EVs	human	--	Rab7a Arf6 flotillin-1	Assessed physical properties of CSF-EVs after severe TBI within 7 days and their proteins associated with neuroregeneration	Kuharic et al., 2019 [112]

HCD-Exos/EVs, humoral cell-derived exosomes/extracellular vesicles; PBCD-Exos, peripheral blood cell-derived Exos; PCD-Exos, plasma cell-derived Exos; PCD-EVs, plasma cell-derived EVs; PBCD-EVs, brain cell-derived EVs in plasma; hbECFCD-Exos, human umbilical cord blood endothelial colony-forming cell-derived Exos; PECD-MVs, blood plasma endothelial cell-derived microvesicles; CSFCD-Exos/MVs, cerebral spinal fluid cell-derived Exos and MVs; CSFCD-EVs, cerebral spinal fluid cell-derived EVs.

**Table 7 jcm-11-03223-t007:** Other cell-derived Exos/EVs and exosome-derived-miRNA studies in TBI.

Origin	Model Species	Involved miRNAs	Other Molecules	Biological Functions/Findings	Study
MPD-EVs BVEC-EVs PD-EVs	mouse	--	--	Increase CNS/hepatic leukocyte recruitment; exacerbate the CNS injury	Hazelton et al., 2018 [116]
SD-EVs	human	--	CDC2, CSNK1A1 CTSD, et al.	Find potential biomarkers to detect TBI by the profiling of SD-EVs	Cheng et al., 2019 [119]
SD-EVs	human	--	MAPK ALOX5 Rap1, et al.	Find promising inflammatory biomarkers to detect TBI by the profiling of SD-EVs	Matuk et al., 2021 [120]
BM-EVs	mouse	--	NF-κB	Activate osteoclast differentiation; inhibit bone loss and fracture rates	Singleton et al., 2019 [121]

MPD-EVs, macrophage/monocyte population-derived extracellular vesicles; BVEC-EVs, brain vascular endothelial cell-derived EVs; PD-EVs, plasma-derived EVs; SD-EVs, saliva-derived EVs; BM-EVs, bone marrow-derived extracellular vesicles.

**Table 8 jcm-11-03223-t008:** Brain-derived Exos/EVs and BD-Exos/EVs-miRNAs studies in TBI.

Origin	Model Species	Involved miRNAs	Other Molecules	Biological Functions/Findings	Study
BD-EVs	mouse	miR-21 miR-212 miR-146, et al.	--	Find promising biomarkers and potential BD-EVs cargoes for TBI	Harrison et al., 2016 [122]
BD-Exos	rat	--	CX43 ERK	Find the biomarkers that promote hippocampal BD-Exos release	Chen et al., 2018 [123]
BD-EVs	mouse	miR-883a3p miR-3057-5p miR-6980-3p et al.	cAMP et al.	Find promising circRNA-miRNA network biomarkers and potential signaling pathways after TBI	Zhao et al., 2018 [124]
BD-EVs	human	--	p-tau PLXNA4 SNAP-25 UBA1, et al.	Find potential monitoring biomarkers and functionally interacting molecules in BD-EVs of CTE after TBI	Muraoka et al., 2021 [125]
BD-Exos	mouse	--	tau p-tau	Identify that BD-Exos could exacerbate motor and cognitive impairments	Wang et al., 2018 [126]
BD-Exos	mouse	--	p-tau TLR-4 p-STAT3	Identify that the inhibition of BD-EV release could alleviate cognitive impairment	Hu et al., 2019 [127]
BD-MVs	mouse	--	lactadherin	Identify that lactadherin could promote BD-MV clearance and improve coagulopathy and the survival of severe TBI	Zhou et al., 2018 [128]
BD-EVs	mouse	--	anionic phospholipids	Identify that anticoagulation targeting membrane-bound anionic phospholipids could improve outcomes of TBI	Dong et al., 2021 [129]

BD-EVs, brain-derived extracellular vesicles; BD-Exos, brain-derived exosomes; BD-MVs, brain-derived microvesicles; circRNAs, circular ribonucleic acids; CTE, chronic traumatic encephalopathy.

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
