# Peer review of "Cell-Derived Exosomes as Therapeutic Strategies and Exosome-Derived microRNAs as Biomarkers for Traumatic Brain Injury"

_jcm, 2022, doi:10.3390/jcm11113223_

Round 1
Reviewer 1 Report
This is an utterly detailed description of cell derived and exosome-derived microsomes, with so many details and cross-references that it was quite a difficult job to concentrate on the real message of the paper.
The use of English language is good, but sentences starting with "And" are disturbing.
In Line 114 I could not really understand the first sentence. Is it possible that it is the title of section 2.2?
Line 175: could not interpret the sentence starting with "As the complement..."
Regarding the tables, it is a bit difficult to understand "Find" as a biological function. It is rather the aim of the study.
Line 327: "but researchers have not been studied in depth" is possibly "researchers have not studied in depth".Line 378: "therapy might exist in a wide range.."
Line 402: The claim is missing from the sentence.
Line 513: "are" is suggested instead of "is" and "maintain" instead of "maintains"
Line 540: "Other cell-derived" is suggested
Line 542: "allegations" is probably not the exact word.
Line 607: what did the authors mean by "insufficient payload"?
I also suggest replacing "et al." in the tables with "etc" - according to Merriam_webster "et al" typically stands in for two or more names, especially in bibliographical information.
Author Response
Response to Reviewer 1 Comments
Dear reviewer:
Sincerely thank you for taking time out of your busy schedule to read our manuscript and give us these useful suggestions. We will respond to your comments point by point as follows:
Point 1: The use of English language is good, but sentences starting with "And" are disturbing.
Response 1: We apologize for the inappropriate use of the language in our manuscript. We have now worked on both language and readability and have also involved native English speakers for language correlations. We really hope that the flow and language level have been substantially improved.
Point 2: In Line 114 I could not really understand the first sentence. Is it possible that it is the title of section 2.2?
Response 2: The sentence is ambiguous, it is not the title of section 2.2. We modify it as “Increasing studies have shown that MD-Exos could mediate microglia-neuron interactions after TBI”.(Line 116-117)
Point 3: Line 175: could not interpret the sentence starting with "As the complement..."
Response 3: The sentence is ambiguous, we modify it as “As the cell-derived proteins, AD-Exos are considered at significantly higher levels than those Exos derived from neurons and other glial cells in CNS”. (Line 176-178)
Point 4: Regarding the tables, it is a bit difficult to understand "Find" as a biological function. It is rather the aim of the study.
Response 4: We agree with this suggestion and have modified the word “Biological function” in all tables throughout the text as “Biological functions / Findings”.
Point 5: Line 327: "but researchers have not been studied in depth" is possibly "researchers have not studied in depth".Line 378: "therapy might exist in a wide range.."
Response 5:
Line 327: "but researchers have not been studied in depth" , the full sentence is “Otherwise, miR-211-3p, miR-188-5p, and miR-465-5p may also be involved in micro-glial autophagy in this study, but researchers have not been studied in depth[79]”. We would like to indicate that the authors of reference 79 did not go into further study.
Line 378: "therapy might exist in a wide range..", the full sentence is “A further study confirmed that BM-MSCs-Exos therapy might exist a wide range of effective doses for the treatment of TBI with at least 7 days of the therapeutic window post-injury [87]”. We would like to indicate that the authors of reference 87 did go into further study.
Point 6: Line 402: The claim is missing from the sentence.
Response 6: We thank the reviewer for pointing out this issue. We add a sentence at the beginning of the paragraph as the claim: “Adipose-derived stem cells (ADSC-Exos) Adsc-exos might play beneficial biological functions after TBI”. (Line 393)
Point 7: Line 513: "are" is suggested instead of "is" and "maintain" instead of "maintains"
Response 7: We agree with this suggestion and have modified the two words. (Line 504)
Point 8: Line 540: "Other cell-derived" is suggested
Response 8: We agree with this suggestion and have used “Other cell-derived” instead of “Other cells-derived”. (Line 529)
Point 9: Line 542: "allegations" is probably not the exact word.
Response 9: We agree with this suggestion and have used “views” instead of “allegations”. (Line 531)
Point 10: Line 607: what did the authors mean by "insufficient payload"?
Response 10: Exosome-based drug delivery is a potential treatment strategy for TBI, but the capacity of drug loading and targeted drug delivery of Exos are limited, which means "insufficient payload" in our opinion. (Line 595)
Point 11: I also suggest replacing "et al." in the tables with "etc" - according to Merriam_webster "et al" typically stands in for two or more names, especially in bibliographical information.
Response 11: Though both of these abbreviations end lists, “et al.” refers to a list of people, whereas “etc” refers to a list of things. So we politely disagree to replace "et al." in the tables with “etc”.
Thank you again for reviewing our manuscript!
Sincerely yours
All authors of this manuscript
Reviewer 2 Report
This topic is very interesting and current. Paper is long, but well structured. Look at some points to improve:
- Lines 53-55: "Exos are secreted by most cell types not only in CNS but also in the peripheral system, including microglia, neurons, astrocytes, mesenchymal stem cells, plasma, cerebrospinal fluid... " What did authors mean for "not only in CNS"? Revise.
- Lines 35-42: "overload, and neuroinflammation, leading to further deterioration.. ". In this introduction section, please report that TBI has high mortality (doi: 10.25259/SNI_697_2020 , doi: 10.3171/2016.11.JNS1618 ) and that its is often associated with cervical trauma ( doi: 10.1097/BRS.0000000000003873, doi: 10.1097/00005373-19760401-00010).
- In the introduction section, please add what is the aim of this review and what authors want to add new to the literature.
- Lines 151-154: "Therefore, miR-5121, miR-124-3p, and miR-155 from MD-Exos or MD-MVs could be a novel therapeutic target for interventions of neurodegeneration and neuronal inflammation after TBI, and miRNA-manipulated MD-Exos may provide a promising therapy for TBI (Table 1)". How did authors select these miRNAs? what kind of literature review was done?
- Lines 177-179: Authors wrote "Numerous studies have demonstrated that Exos are closely related to TBI [15].. " however only 1 reference was reported. Revise.
- At the moment, according to the authors, at what point in the chain it is possible to act in favor of the patient. Please report before conclusion.
- The lack of precise endpoints and of a diagram flow in the search represent limitations of the paper. Discuss about these points. The topic is very broad and therefore it appears difficult to carry out a standard-oriented research.
Author Response
Response to Reviewer 2 Comments
Dear reviewer:
Sincerely thank you for taking time out of your busy schedule to read our manuscript and give us these useful suggestions. We will respond to your comments point by point as follows:
Point 1: Lines 53-55: "Exos are secreted by most cell types not only in CNS but also in the peripheral system, including microglia, neurons, astrocytes, mesenchymal stem cells, plasma, cerebrospinal fluid... " What did authors mean for "not only in CNS"? Revise.
Response 1: We thank the reviewer for pointing out this issue. We have modified the sentence as “Exos are secreted by most cell types in the central nervous system (CNS) and the pe-ripheral system, including……”.(Line 53-54)
Point 2: Lines 35-42: "overload, and neuroinflammation, leading to further deterioration.. ". In this introduction section, please report that TBI has high mortality (doi: 10.25259/SNI_697_2020 , doi: 10.3171/2016.11.JNS1618 ) and that its is often associated with cervical trauma ( doi: 10.1097/BRS.0000000000003873, doi: 10.1097/00005373-19760401-00010).
Response 2: We appreciate the reviewer for this meaningful advice. We are very willing to add the content you suggested in the introduction section and we have found two articles (doi: 10.25259/SNI_697_2020 , doi: 10.1097/BRS.0000000000003873 ), but we have not found the other two articles you suggested (doi: 10.3171/2016.11.JNS1618, doi: 10.1097/00005373-19760401-00010) in PubMed and other literature search sites, could you please provide us with more details about these two articles and we will add the content you suggested in the next revision of the manuscript.
Point 3: In the introduction section, please add what is the aim of this review and what authors want to add new to the literature.
Response 3: We thank the reviewer for this meaningful advice. We have already added the content you suggested in the last paragraph of the introduction section. “Therefore, the aim of this review is to analyze Exos and their miRNA cargoes as treatment strategies after TBI, to provide new strategies for preventing the long-term progression of the disease, and to provide new literature for the diagnosis and treatment of TBI by cell-derived Exos.” (Line 91-94)
Point 4: Lines 151-154: "Therefore, miR-5121, miR-124-3p, and miR-155 from MD-Exos or MD-MVs could be a novel therapeutic target for interventions of neurodegeneration and neuronal inflammation after TBI, and miRNA-manipulated MD-Exos may provide a promising therapy for TBI (Table 1)". How did authors select these miRNAs? what kind of literature review was done?
Response 4: We summarize the findings of references 30-34 and come to the conclusion that “miR-5121, miR-124-3p, and miR-155 from MD-Exos or MD-MVs can be novel therapeutic targets for interventions of neurodegeneration and neuronal inflammation after TBI. (Line 153-156)
Point 5: Lines 177-179: Authors wrote "Numerous studies have demonstrated that Exos are closely related to TBI [15].. " however only 1 reference was reported. Revise.
Response 5: We thank the reviewer for pointing out this issue. We have added other references at this location. (Line 178-179)
Point 6: At the moment, according to the authors, at what point in the chain it is possible to act in favor of the patient. Please report before conclusion.
Response 6: We thank the reviewer for this meaningful advice. We have reported this topic now. “it shows that some neural cells, especially neural stem cells, astrocytes and microglia, and their miRNAs such as miR-124, miR-141 and miR-32, etc. may become promising strategies for targeted drug delivery in the treatment of TBI.” (Line 626-628)
Point 7: The lack of precise endpoints and of a diagram flow in the search represent limitations of the paper. Discuss about these points. The topic is very broad and therefore it appears difficult to carry out a standard-oriented research.
Response 7: We thank the reviewer for these meaningful advices. We have added the “conclusion” at the end of the article and discussed the limitaition of this review. In addition, since each reference in 2.1-2.8 is presented in table1-table8, we did not make the diagram flow. If you still think it is necessary to make a diagram flow after reading our response, we will add it in the next revision of the manuscript. We once again sincerely thank the reviewer for reading our manuscript! (Line 626-631)
Thank you again for reviewing our manuscript!
Sincerely yours
All authors of this manuscript
Reviewer 3 Report
The review is well-structured and broadly covering the field of EVs and TBI
Remarks:
Line 49-50. “to remove toxic proteins or aggregate to form the cytoplasm of cells” makes no sense
Legend figure 1: plasma is not a cell type.
Line 114: “MD-Exos mediated mechanism of microglia-neuron interaction after TBI” sentence is incomplete and superfluous.
Table 1: Kumar et al., 2017 (34) should be Kumar et al., 2017 (6)
Line 160 astrocytes do not form the BBB!
Lin 172 “with neurotrophic and neuroprotective” ?
Line 373-377. Should be rephrased
Line 406 replace driven, it is past participle.
Line 424 section 2.6 should be rewritten because the use of the word humoral in this context is confusing. Exosomes are derived from cells and they can be present in bodily fluids.
Line 542 replace allegations
Author Response
Response to Reviewer 3 Comments
Dear reviewer:
Sincerely thank you for taking time out of your busy schedule to read our manuscript and give us these useful suggestions. We will respond to your comments point by point as follows:
Point 1: Line 49-50. “to remove toxic proteins or aggregate to form the cytoplasm of cells” makes no sense.
Response 1: We agree with this suggestion and have modified the sentence as “on the other hand, they may be released from cells and send biological messages to other cells.”(Line 50-51)
Point 2: Legend figure 1: plasma is not a cell type.
Response 2: We have modified the first sentence of figure 1 legend as “Exosomes are secreted by most cell types not only in CNS but also in the peripheral system, including microglia, astrocytes, neurons, mesenchymal stem cells (MSCs), and peripheral blood cells”.
Point 3: Line 114: “MD-Exos mediated mechanism of microglia-neuron interaction after TBI” sentence is incomplete and superfluous.
Response 3: The sentence is ambiguous. We modify it as “Increasing studies have shown that MD-Exos could mediate microglia-neuron interac-tions after TBI”. (Line 116-117)
We apologize for the inappropriate use of the language in our manuscript. We have now worked on both language and readability and have also involved the English editor of MDPI and native English speakers for language correlations. We really hope that the flow and language level have been substantially improved.
Point 4: Table 1: Kumar et al., 2017 (34) should be Kumar et al., 2017 (6).
Response 4: It was indeed our mistake, and we have corrected it now.
Point 5: Line 160 astrocytes do not form the BBB!
Response 5: We thank the reviewer for pointing out this issue. We have used “maintaining the BBB” instead of “maintaining and forming the BBB”. (Line 161)
Point 6: Lin 172 “with neurotrophic and neuroprotective” ?
Response 6: The sentence is incomplete. We modify it as “Interestingly, AD-Exos and AD-EVs secreted in normal conditions are known to be enriched with neurotrophic and neuroprotective effects”. (Line 171-173)
Point 7: Line 373-377. Should be rephrased.
Response 7: We rephrased the sentence “In addition, intravenous injection of 2-dimensional (2D) conventional conditions or 3-dimensional (3D) collagen scaffolds cultured BM-MSCs-Exos could also achieve a similar effect in rats after TBI, and the BM-MSCs-Exos cultured in 3D conditions provide better outcomes in spatial learning compared to Exos from 2D culture” as
“In addition, BM-MSCs-Exos cultured with collagen scaffolds could also achieve similar effects in improving functional recovery of TBI rats”. (Line 367-369)
Point 8: Line 406 replace driven, it is past participle.
Response 8: We apologize for the inappropriate use of the language in our manuscript. We have used “drove” instead of “driven”. (Line 398)
Point 9: Line 424 section 2.6 should be rewritten because the use of the word humoral in this context is confusing. Exosomes are derived from cells and they can be present in bodily fluids.
Response 9: We apologize for the confusion generated by the previous version of the manuscript and sincerely hope that our logic is now easier to follow with this new version for section 2.6. We have revised the title of this section as ” Humoral cells-derived exosomes and exosome-derived microRNAs in TBI”, and the content of this section has also been modified accordingly.
Point 10: Line 542 replace allegations.
Response 10: We agree with this suggestion and have used “views” instead of “allegations”. (Line 530)
Thank you again for reviewing our manuscript!
Sincerely yours
All authors of this manuscript
Round 2
Reviewer 2 Report
Authors revised very well the manuscript and solved all my criticisms.
Well done.
Paper suggested to discuss are:
- Posttraumatic synchronous double acute epidural hematomas: Two craniotomies, single skin incision. Surg Neurol Int. 2020 Dec 11;11:435.
- Traumatic brain injury: classification, models, and markers. Biochem Cell Biol. 2018 Aug;96(4):391-406.
- Establishing the Injury Severity of Subaxial Cervical Spine Trauma: Validating the Hierarchical Nature of the AO Spine Subaxial Cervical Spine Injury Classification System. Spine (Phila Pa 1976). 2021 May 15;46(10):649-657.
- An update on diagnostic and prognostic biomarkers for traumatic brain injury. Expert Rev Mol Diagn. 2018 Feb;18(2):165-180.